# Exploiting Patch Sizes and Resolutions for Multi-Scale Deep Learning in Mammogram Image Classification

**DOI:** 10.3390/bioengineering10050534

**Published:** 2023-04-27

**Authors:** Gonzalo Iñaki Quintana, Zhijin Li, Laurence Vancamberg, Mathilde Mougeot, Agnès Desolneux, Serge Muller

**Affiliations:** 1GE HealthCare, 283 Rue de la Minière, 78530 Buc, France; 2ENS Paris-Saclay, Centre Borelli, 91190 Gif-sur-Yvette, France

**Keywords:** breast imaging, artificial intelligence, deep learning, computer aided detection or diagnosis (CAD), convolutional neural networks (CNNs), multi-scale classification

## Abstract

Recent progress in deep learning (DL) has revived the interest on DL-based computer aided detection or diagnosis (CAD) systems for breast cancer screening. Patch-based approaches are one of the main state-of-the-art techniques for 2D mammogram image classification, but they are intrinsically limited by the choice of patch size, as there is no unique patch size that is adapted to all lesion sizes. In addition, the impact of input image resolution on performance is not yet fully understood. In this work, we study the impact of patch size and image resolution on the classifier performance for 2D mammograms. To leverage the advantages of different patch sizes and resolutions, a multi patch-size classifier and a multi-resolution classifier are proposed. These new architectures perform multi-scale classification by combining different patch sizes and input image resolutions. The AUC is increased by 3% on the public CBIS-DDSM dataset and by 5% on an internal dataset. Compared with a baseline single patch size and single resolution classifier, our multi-scale classifier reaches an AUC of 0.809 and 0.722 in each dataset.

## 1. Introduction

Breast cancer is one of the most common cancers among women in North America and Europe. In 2020, female breast cancer was the most frequently diagnosed cancer: it accounted for 11.7% of all cancer cases (with an estimated 2.3 million cases), and 6.9% of all cancer deaths worldwide [1]. Studies have shown that early detection is key to improve breast cancer survival rates, as in those cases, the patient can be cured in nine out of ten cases [2]. Imaging techniques play an important role in breast cancer detection, diagnosis, and follow-up therapy. Full field digital mammography (FFDM), commonly known as 2D mammography, is a two-dimensional (2D) X-ray imaging technique of the breast that is widely acknowledged to be effective in reducing breast cancer mortality over an asymptomatic population [3]. It is also the primary imaging technique for breast cancer screening. Despite this, the accuracy of FFDM can vary significantly depending on breast density, ranging from 47.8% for dense breasts to 98% for less dense breasts, with an average of 77.6% [4]. This is mainly due to breast tissue overlapping, and this directly impacts screening, as it can decrease the lesion visibility in dense breasts. In addition, breast tissue overlapping can also create false signs that resemble radiological findings, which results in unnecessary recalls [5]. In today’s breast cancer screening with mammography, the average recall rate varies from 5% to 12%, while less than 5% of recalled patients are diagnosed with breast cancer [6]. Improving the efficiency of mammography breast cancer screening is needed in clinical practice.

Computer aided detection or diagnosis (CAD) systems are designed to assist clinicians in detecting or classifying potential abnormalities in clinical exams. Over the past two decades, CAD systems have been extensively developed for breast cancer detection and diagnosis to improve the efficiency of screening and diagnostic mammography [7]. They can be used to highlight suspicious lesions in 2D mammograms, helping clinicians reduce reading time [7], or acting as a clinical support to indicate the likelihood of breast cancer presence, or to triage normal and abnormal exams in population based screening programs to improve image reading efficiency. Traditional CAD systems typically rely on hand-crafted features, a combination of expert knowledge and mathematical models from image processing, pattern recognition, and classic machine learning algorithms. To date, there is no clinical evidence supporting an increased breast cancer detection performance using traditional CAD algorithms in screening mammography. Studies have shown that the use of traditional CAD systems in clinical practice allows for limited to no increase in sensitivity, while introducing a non-negligible increase in the recall rate in most cases [7,8,9,10]. This is mainly attributed to the fact that traditional CAD algorithms need a relatively large number of markers to guarantee a high detection sensitivity, which leads to high false positive rates [7] and high recall rates when used in clinical practice.

Recently, with the growing success of artificial intelligence, particularly deep learning (DL), there has been a noticeable trend towards creating CAD systems that rely on DL technology [9]. DL-based CAD replaces hand-crafted features by representation learning using deep neural networks trained in a data-driven fashion. It has the potential to address the issue of the increased recall rate in traditional CAD systems. This has been demonstrated in several studies [11,12,13,14,15,16,17], that show a comparable or increased breast cancer detection performance in terms of sensitivity and specificity of DL-based CAD compared to an average radiologist under different clinical set-ups. Although the translation of the results from these studies to clinical practice is still under active investigation, DL-based CAD is effectively considered as a promising technology to improve breast cancer screening with 2D mammography [9].

### 1.1. 2D Mammography Image Classification

In this work, we aim to design a binary classifier for 2D mammograms and to address some of its open questions, including the impact of the patch size and input image resolution on performance. The algorithm predicts a cancer score that indicates the likelihood of the presence of cancer, which can then be thresholded to obtain a binary prediction (cancer/no cancer).

In the literature, two main approaches for classifying 2D mammograms can be distinguished: fully image-based and patch-based. Fully image-based methods typically consist of a deep convolutional neural network (CNN) that takes an entire mammogram as input, and outputs a cancer score. Fully image-based classifiers are the straightforward application of DL models used for natural images, and they only exploit image-level annotations (i.e., ground-truth labels given to the entire image). As explained by Shen et al. [18], mammograms have the particularity in which the regions that contain the lesions usually represent a small part of the image (i.e., 0.08% of the total image), and this determines the class of the entire image [18]. This “needle in a haystack” problem makes full image classifiers hard to train, as they have to be trained on large datasets in order to reach a good performance.

Moreover, patch-based approaches [18,19] directly exploit lesion-level annotations by first training a CNN-based patch classifier and then extending it to a whole image classifier. The first method used to extend the patch classifier is to apply it to the entire mammogram in a sliding window fashion, and combine the predictions of a subset of selected patches via voting or other aggregation techniques [20,21,22,23]. However, by processing each patch independently, these approaches do not exploit the locality information between the patches, and they are computationally inefficient as convolutions in the overlapping region of two patches are computed twice. Another way to extend a patch classifier into a whole image classifier consists in applying the patch classifier to the mammogram, and then appending and fine-tuning the additional convolutional layers that combine the features from the different regions [18,19]. This approach allows to efficiently train whole image classifiers that incorporate lesion-level annotations, while avoiding the drawbacks of the sliding window approach.

The two families of methods described above intend to classify a single mammogram. However, a mammography screening study typically consists of four images: two clinical views, craniocaudal (CC) and mediolateral oblique (MLO), and two lateral views, for the left and right breasts. Multi-view classifiers are currently the state-of-the-art in 2D mammogram classification, as they exploit the multi-view nature of mammography studies, by leveraging bilateral asymmetries and multiple views of the lesions. Multi-view classification is typically achieved by aggregating the predictions of the independently processed views, either by concatenating the feature vectors extracted from the different views and applying a linear layer [24], or by concatenating the feature maps from each view and appending additional convolutional layers [15,18,19,25]. Recent approaches propose attention mechanisms for feature transfer between the clinical and lateral views [25,26,27].

Several questions remain open in mammogram classification, such as the impact of the patch size, and of the spatial resolution of the mammograms. While the former only concerns patch-based methods, the latter is of interest to all types of classifiers. Patch-based methods are limited by the patch-size they are based on, due to the multi-scale nature of breast lesions. Breast lesions span over a wide range of sizes, and can range from a few millimetres to a few centimetres. On the one hand, small patches are good at detecting small lesions, but they cannot entirely contain large lesions, which might affect the final performance. On the other hand, sufficiently large patches ensure that large lesions are fully enclosed in the patch, but this may adversely affect the detection of small lesions. This is because the CNN may fail to capture the fine details and subtle features of small lesions. In the deep learning literature, we find several multi-scale approaches to classify, detect, and segment objects of different sizes, such as the inception block [28], dilated convolutions [29], and feature pyramid networks (FPN) [30]. The inception block aims to capture information at multiple scales within the same convolutional layer by implementing different filter sizes in each block. Dilated convolutions contain gaps between the values of the filter, which increase the effective receptive field of the layer, and can be used for capturing multi-scale information by adjusting the dilation rate. Finally, feature pyramid networks generate feature maps at multiple levels of the convolutional backbone, that contains information at different scales. FPN implements bottom-up and top-down pathways that enhance the features at different scales, and allow for the flow of semantic information from deep features to more shallow ones. Although first developed for object detection, FPN has been recently extended to classification problems [31,32]. These classic DL multi-scale approaches can help to alleviate the effect of the patch size, but are intrinsically limited as they cannot make large lesions fit in small patches.

The impact of decreasing the resolution of mammograms is an open question of interest for all types of classifiers. Several studies resize mammograms when training DL models [25,26,27,33] as a means of decreasing memory consumption and fitting larger batches in memory. This is a usual practice for natural images but can severely harm the performance for mammograms, as the determinant regions can only cover a small portion of the image. However, we are not aware of a study that analyses the impact of resolution reduction on the performance of this type of image.

### 1.2. Contributions

In this work, we focus on the patch-based approach for whole image classification and aim to address some of its open questions. As described in Section 1, patch-based classification consists first of training a patch classifier and then extending it to a whole image classifier. The main contributions of this paper are the following:The impact of the patch size is studied, both on the patch classifier and on the whole image classifier. For the patch-classifier, the patch size effect on lesions of different sizes is analysed;The impact of decreasing the input image resolution on the two classifiers is studied, as well as its effect on lesions of different sizes;A multi-patch size and a multi-resolution approach for classifying whole images are proposed, that leverage patch classifiers adapted to different lesion sizes. These multi-scale models are shown to outperform single patch-sized, single resolution classifiers.

Multi-view classification is beyond the scope of this research.

## 2. Materials and Methods

### 2.1. Datasets

Two 2D mammography datasets were considered: CBIS-DDSM and an internal GE HealthCare (GEHC) FFDM dataset. The two datasets contain lesion-level annotations, that enable to extract patches for training the patch classifier.

CBIS-DDSM [34] is the largest public curated dataset of 2D scanned film mammograms. It contains 1566 cases, 758 of which are biopsy-proven cancers. The official test split was used as the validation set, which left a total of 1293 cases for training and 351 cases for validation. The images were downsampled to a resolution of 100 μm/pixel, which matched the resolution of GEHC images. The original resolution of CBIS-DDSM images was around 50 μm/pixel, but this varied due to different scanners used to digitize the films.

The GEHC dataset contains 1539 cases, of which 363 are biopsy-proven cancers and 351 contain benign biopsied lesions. The remaining 747 are normal cases, which are the studies in which no suspicious lesion was found in the breasts, and no biopsy was conducted. All normal cases were confirmed by a follow-up exam. The anonymized data were collected from a single institution following the EU General Data Protection Regulation. The dataset was split in into training (1237 cases), validation (201 cases), and testing (101 cases) sets in a stratified fashion, which took into account the case pathology (benign or malignant), the lesions contained in the image (mass and calcification), and the description or sub-type of the lesions (e.g., spiculated mass, oval mass, granular calcification, etc.).

### 2.2. Patch Extraction

Ten normal or background patches, and at least ten lesion patches were extracted from each image that contained a lesion (mass or calcification), with two different strategies: “fixed” and “random” extractions. For every lesion, a “fixed” patch centred on the lesion was extracted. If the lesion was too large to be entirely contained in the patch, the space covered by the lesion was divided into a grid of N×M non-overlapping patches, which were then incorporated into the patch dataset. This assured that every part of the lesion was represented in the dataset, but may have introduced an undesirable bias, as most patches coming from large lesions contain the lesion fragment in the corners. To reduce this bias, the patch dataset was enriched with “random” lesion patches, centred on random positions of the lesion. The extracted patches had an intersection over union (IoU) smaller than 0.5 between each other, to avoid generating patches that were too similar. We remark that there exists a strong class imbalance in the patch dataset, as normal patches are extracted from every image, but most images contain only one lesion.

### 2.3. Patch Classifier

As in [18,19], fixed-sized patches were classified into 5 classes: background or normal, benign calcification, malignant calcification, benign mass, and malignant mass. To set the classifier’s architecture, networks from the DenseNet [35] and ResNet [36] technologies were benchmarked. We decided to use DenseNet-121, as it achieves the highest performance while being the deepest and smallest model. DenseNet implements feature concatenation, which makes them more parameter-efficient than ResNet. For instance, DenseNet-121 has $7\times 10^{6}$ parameters and 121 layers, while ResNet-34 has $21.3\times 10^{6}$ parameters and only 34 layers. This makes DenseNet less prone to overfitting and capable of learning more complex representations. A complete study on the network architecture was nevertheless beyond the scope of this research.

To analyse the impact of the patch size, the patch classifier was trained with three different patch sizes: 256, 512, and 768 pixels, all at a resolution of 100 μm/pixel. Additionally, to analyse the impact of the input resolution, the patch classifier was trained at three different resolutions: 100, 150, and 200 μm/pixel, while maintaining a fixed patch-size of 512 pixels. For the CBIS-DDSM dataset, the weights were initialized from ImageNet and optimized with the stochastic gradient descent (SGD) and a cosine annealing learning rate scheduler [37]. For the GEHC dataset, the weights were initialized from the CBIS-DDSM model and the last two dense layers were fine-tuned. Classic data augmentation (flips and rotations) was performed, and lesion patches were oversampled during the training to tackle the class imbalance mentioned in Section 2.2. To estimate the error bars, each model was trained 10 times.

For better characterizing the impact of the patch size and of the resolution, the lesions were divided into three size groups: small lesions (the maximal dimension was smaller than 25.6 mm, or 256 pixels at 100 μm), medium lesions (the maximal dimension was between 25.6 and 51.2 mm, or 256 and 512 pixels at 100 μm), and large lesions (the maximal dimension was larger than 51.2 mm, or 512 pixels at 100 μm). The performance was analysed for each lesion size and type (calcification or mass) for a total of six groups: small, medium, and large calcification; small, medium, and large mass. The patch classifiers were evaluated in each of the lesion groups using two AUC-based metrics: normal vs. abnormal AUC, and benign vs. malignant AUC. Normal vs. abnormal AUC seeks to characterize the ability of the classifier to find or detect a given lesion type and size (e.g., medium mass). It was obtained by evaluating the model in a subset of patches that only contained lesions of that type and size and normal patches in equal proportions. The benign vs. malignant AUC aims to evaluate the ability of the classifier to tell if a lesion of a given size and type is benign or malignant. It was obtained by evaluating the model only on patches with that particular type and size of lesion. Throughout the remainder of the article, patch and lesion sizes are expressed in terms of the number of pixels at a 100 μm/pixel resolution, e.g., 256, 512, and 768 pixels. The results are discusseed in Section 3.1.

### 2.4. Base Whole Image Classifier

The patch classifier of Section 2.3 was extended to a whole image binary classifier (cancer/no cancer) by removing the global average pooling (GAP) and fully connected (FC) layers, and appending a convolutional block (see Figure 1). The previously trained patch classifier was used as a feature extractor and was applied to the entire image, which gave feature maps that were proportionally larger in the spatial dimensions. For instance, our DenseNet-121 patch classifier had a spatial reduction factor of 32, and output feature maps of 16×16×1024 size when the input patches were of 512×512 pixels. When applied to entire mammograms of 2850×2394 size, it output feature maps of 89×75×1024 size. The convolutional block was then applied to the feature map to aggregate and combine the patch-level predictions into another feature map of reduced spatial dimensions, that maintained the number of channels. In this paper, the bottleneck residual block proposed in [18] was used, as it efficiently implements skipped connections to avoid vanishing gradients [36]. GAP and FC layers were appended after the reduced feature map to perform the final binary classification.

We want to stress that the size of the feature maps depends only on the input image size and on the reduction factor, which is architecture-dependent. As a consequence, the patch classifiers trained on patches of different sizes output feature maps of equal size when applied to an entire mammogram.

One whole image classifier was trained for each of the patch classifiers of Section 2.3, and obtained 5 base whole image classifiers: patch sizes 256, 512, and 768 at 100 μm resolution; and resolutions 150 μm and 200 μm with a patch size of 512 pixels. To estimate the error, each classifier was trained 5 times. The mean performance in terms of AUC, specificity (SP), and accuracy (Acc), as well as the standard deviation and the significance level (*p*-value) were calculated. The *p*-values were obtained by comparing each classifier to the best performing base classifier of each experiment, using the one-sided Welch’s *t*-test with unknown variances. The best performing base classifiers used in the *t*-test were: resolution 100 μm for the CBIS-DDSM and GEHC datasets in the resolution experiment, patch size 256 for CBIS-DDSM in the patch size experiment, and patch size 512 for the GEHC dataset in the patch size experiment. We remark that, in contrast to the AUC, which is a metric computed on the entire ROC curve, the specificity and accuracy are relative to one particular operating point. While the choice of the operating point typically depends on the application (e.g., cancer prediction, normal exam triage, etc.), in this work, it is fixed to have a sensitivity (Se) of 0.75 (ref. [38] defines the acceptable range in sensitivity as starting at 0.75).

### 2.5. Multi-Resolution & Multi-Patch Size Whole Image Classifier

As will be shown in Section 3.1, there is not one unique patch size or resolution that outperforms for all of the lesion groups. Inspired by this result, we propose a multi-resolution classifier and a multi-patch size classifier that leverages patch classifiers adapted to different lesion sizes (Figure 2).

The multi-patch size classifier contains three whole image classifiers based on three different patch sizes (256, 512, and 768 pixels). These classifiers were applied to the input image and the three output feature vectors, that were then concatenated and forwarded through a multi layer perceptron (MLP) that gave the final classification (see Figure 2). For the multi-resolution classifier, the input image was downsampled two times, and three resolution specific classifiers were applied to each version of the image (resolution 100, 150, and 200 μm). The features were also concatenated and forwarded through an MLP. Please note that in the case of the multi-resolution classifier, the feature maps extracted by each base classifier (before the GAP layer) have different spatial dimensions, as the image has been resized.

In addition, the proposed multi-scale models (multi-patch size and multi-resolution) were compared with a feature pyramid network (FPN) classifier, based also on DenseNet-121. The FPN whole image classifier was constructed by first training an FPN patch classifier that combines features from three different spatial resolutions for 5-class multi-scale classification. The FPN-based patch classifier was then extended to a whole image classifier by appending residual blocks and concatenating the features, as was carried out for the multi-patch size and multi-resolution classifiers. The patch size of the FPN classifier was set to 768 pixels, which is the largest patch size used in this work and allows it to detect large lesions. The detection and classification of small lesions was assured by the multi-scale nature of the FPN. The choice of FPN patch size was also validated experimentally, and FPN classifiers based on 256 and 512 patch sizes were found to yield a lower performance.

## 3. Results

### 3.1. Patch Classifier

Figure 3a,b shows the two metrics introduced in Section 2.3 for the patch classifiers with different patch sizes. Figure 4a,b shows the same two metrics for the patch classifiers at different resolutions. We also include the level of statistical significance of the one-sided Welch’s *t*-test, with respect to the 256 size classifier (Figure 3a,b), and to the 100 μm classifier (Figure 4a,b).

Figure 3a,b shows that small lesions are better classified when small or medium patches are used (sizes 256 and 512). When the lesion size increases, the lesions stop being entirely contained in the patches, and the performance of small patch size classifiers drops. In contrast, patch classifiers that utilize larger patch sizes, such as 768 pixels, demonstrate superior performance when classifying large lesions. The fact that in some cases, medium-sized patch classifiers outperform for small lesions, and large-sized patch classifiers outperform for medium lesions, it suggests that context information in the patch (i.e., breast tissue surrounding the lesions) may be relevant for the final classification. In Figure 4a,b, we can observe an analogous behaviour: decreasing the resolution while maintaining the fixed patch size allows for larger lesions to be fully contained in the patches, which increases the performance for larger lesions. As a consequence, low resolution patch classifiers (e.g., 150 and 200 μm) perform poorly for small lesions, but outperform for large lesions. Medium lesions form a transition group in which we cannot clearly distinguish an outperforming patch size or resolution.

This is also illustrated in Figure 5, that shows the class activation maps (CAMs) of the ground-truth class for a medium malignant mass and a small benign calcification of the CBIS-DDSM dataset. The CAMs were extracted from the last convolutional layer using the grad-CAM method [39]. Figure 5a shows the CAMs for the three patch sizes (256, 512, and 768 pixels) and Figure 5b shows the CAMs for the three resolutions (100, 150, and 200 μm). Both Figures also include the ground-truth segmentation of the lesion. In the left column of Figure 5a, we observe that when the lesion is not entirely contained in the patch, which is the case when the patch size is 256, the classifier performs poorly. Increasing the patch size (sizes 512 and 768) enables the lesion to be fully contained in the patch and adds context information (surrounding breast tissue), which increases the classification score and the localization performance of the classifier. We remark that the localization of patch size 768 is more accurate than that of 512 size. On the contrary, when the lesion is small (right column of Figure 5a), increasing the patch size decreases the classification score. In Figure 5b, we see the same effect when analysing the different resolutions: a decrease in the resolution increases the performance for the medium mass (left column), but decreases it for the small calcification.

From Figure 3, Figure 4 and Figure 5, we can conclude that there is not a single patch size or a single resolution that is well adapted to all of the lesion sizes. Combining different patch sizes or resolutions can increase performance for all of the lesion groups, and therefore increase global performance.

### 3.2. Base Whole Image Classifier

Table 1 and Table 2 show the mean AUC, specificity (Sp), and accuracy (Acc), as well as their significance levels for whole image classification in the CBIS-DDSM and GEHC datasets. In CBIS-DDSM, we can see that the base classifier with a patch size of 256 outperforms the other base classifiers with a mean AUC of 0.784. In the GEHC dataset, the base classifier with patch size of 512 outperforms the other base classifiers. We note the non-negligible decrease in performance when reducing the resolution of mammograms. It drops by 0.020 AUC for CBIS-DDSM and by 0.060 AUC for the GEHC dataset, when the resolution decreases from 100 μm to 150 μm (reduction factor of 1.5). When the reduction factor is 2 (resolution decreases from 100 μm to 200 μm), performance is reduced by 0.028 AUC for CBIS-DDSM and by 0.105 AUC for the GEHC dataset. In terms of specificity and accuracy for the given operating point (Se = 0.75) the same trends observed in the AUC are confirmed, but with a generally lower statistical significance (see Table 1 and Table 2).

### 3.3. Multi-Resolution & Multi-Patch Size Whole Image Classifier

The multi-patch size and the multi-resolution classifiers outperform their respective base classifiers in the two considered datasets (see Table 1 and Table 2). The multi-patch size has the highest performance: it attains 0.809 AUC in CBIS-DDSM and 0.722 in the GEHC dataset. All these performance differences are significant at different levels of statistical significance, as indicated in Table 1 and Table 2. The multi-resolution classifier consumes less computational resources than its multi-patch size counterpart: its training time is three times smaller and its total GPU memory consumption is 10% lower. Despite this, the reduction in AUC is of only 0.020 in CBIS-DDSM, and of 0.013 in the GEHC dataset (see Table 1 and Table 2).

In terms of specificity and accuracy, the multi-patch size classifier outperforms the base classifiers in CBIS-DDSM and in the GEHC dataset. In CBIS-DDSM, it has a Sp of 0.710 and an Acc of 0.730 in contrast to 0.656 and 0.703 for the best base classifier. In the GEHC dataset, it has a Sp of 0.552 and an Acc of 0.651, in contrast to 0.487 and 0.619 for the best base classifier. The multi-resolution classifier outperforms the base classifiers in CBIS-DDSM, but fails to outperform the best base classifier in the GEHC dataset. In CBIS-DDSM, it attains a Sp of 0.670 and an Acc of 0.710, in contrast to a Sp of 0.627 and an Acc of 0.689 for the best base classifier. In the GEHC dataset, the two classifiers obtain a Sp and an Acc that cannot be considered to be statistically different.

In Table 1 and Table 2, it can be seen that our multi-patch size classifier significantly outperforms the FPN by 0.021 AUC in CBIS-DDSM and by 0.025 in the GEHC dataset. The multi-resolution classifier has a slightly higher performance than the FPN-based classifier, yielding an AUC difference of 0.001 in CBIS-DDSM and of 0.012 in the GEHC data. However, this performance difference is not confirmed by the statistical test in CBIS-DDSM, which can be due to an insufficient number of independent runs. In terms of specificity and accuracy for the given operating point, the multi-patch size classifier outperforms the FPN in the two datasets; the multi-resolution classifier outperforms the FPN in CBIS-DDSM, but is outperformed in the GEHC dataset, despite this performance difference not being statistically significant.

## 4. Discussion

In CBIS-DDSM, the classification performance of the DL-based classifiers was found to be severely impacted when using the original train-test split with respect to random splits [40]. This can be explained by the fact that the original test set was acquired at a different time and has been reported to contain intrinsically more difficult cases [40]. For instance, the AUC of the patch-based, single-view classifier of Petrini et al. [19] decreased from 0.8757, when using a random split, to 0.8033 when using the original train-test split; the AUC of the single-view model of Wei et al. [40] dropped from 0.91825 to 0.7964. Thus, we compare the performance of our classifiers to those of other models trained and evaluated using the original train-test split from CBIS-DDSM. The AUC of our multi-patch size classifier in CBIS-DDSM is comparable to the highest reported AUC in the CBIS-DDSM official test set for single-view whole image classifiers (0.803 ± 0.010 for Petrini et al. [19], 0.7964 for Wei et al. [40]). In addition, it outperforms the patch-based model of Shen et al. [18] (AUC = 0.75 [19]) and the classifier of Almeida et al (AUC = 0.6829 [41]) in the original train-test split. It also outperforms the single-view transformer-based model of Van Tulder et al. [26], despite its AUC of 0.757 being declared in a supposedly easier random split of CBIS-DDSM. In contrast, all of our single-view classifiers are outperformed by the multi-view models of Petrini et al. and Wei et al., with reported AUCs of 0.8483 and 0.8313. This illustrates the superiority of the multi-view classification discussed in Section 1. As in terms of the AUC, our single-view multi-patch size classifier (Se 0.750 and Sp 0.710) is outperformed by the multi-view model of Petrini et al. [19], with a reported sensitivity and specificity of 0.7568 and 0.7716. The specificity of the multi-patch size classifier in CBIS-DDSM (Sp 0.71 for Se 0.75) matches the minimum specificity of radiologists [38].

Patch-based approaches still have open questions, such as “What is the optimal patch-sampling strategy?” or “What is the best way to combine the features from the patch-classifier?”. There are also still research opportunities in multi-scale classification for mammograms. For example, instead of constructing the multi-resolution classifier by combining three resolution-specific classifiers, we could imagine training a single patch-classifier with patches at various resolutions to then extend it to a multi-resolution whole image classifier. This would enable to reduce training and inference time, and to obtain a patch-classifier that extracts richer representations. In addition, the choice of the patch sizes could be incorporated into the training loop by clustering the images by the size of the lesions they contain and automatically setting the patch sizes, in a similar way to what is achieved in Yolo v5 [42]. This would ease the transferability of this research to other medical imaging modalities that may have a different spectrum of lesion sizes. Finally, patch-based classifiers rely heavily on lesion-level annotations, which are rare and expensive in medical imaging. If the datasets used for training are not large enough, the models can be easily over fit. Unsupervised and semi-supervised learning methods alleviate this issue by incorporating non-annotated data to the training, that are usually available in larger numbers than annotated data, and have the potential to improve the model’s performance and generalization. For instance, refs. [43,44] show that training with labelled and unlabelled images using the MixMatch algorithm [45] improves performance in BIRADS classification, especially when the number of annotated images is small. In addition, purely unsupervised clustering approaches have been applied to radiomics features to produce a reduced and de-noised feature space [46]. These reduced features can then be used to train classification models with smaller annotated datasets, while avoiding overfitting [46]. The impact of dataset size on patch-based classifiers trained in a fully supervised way should be evaluated, and unsupervised learning and semi-supervised learning techniques should be considered.

## 5. Conclusions

The impact of the patch size and input image resolution in patch-based mammogram classifiers has been investigated. It has been shown that there is not a unique patch size or resolution that is optimal for every lesion size and type. To overcome this issue, a multi-patch size and a multi-resolution approach has been proposed for whole image classification. The multi-patch size solution outperformed FPN in the two datasets, and improved the AUC by 0.025 (+3%) on the public CBIS-DDSM dataset, and by 0.034 (+5%) on the GEHC dataset, compared to the best single patch size baseline classifier. The multi-resolution approach gave an increase of 0.025 (+3%) on CBIS-DDSM and of 0.021 (+3%) on the GEHC data, compared to the best single resolution classifier.

While the findings of this study are relevant, it has some limitations that need to be addressed in future research. A complete and in-depth study on patch sampling strategies and their impact on mammogram classification should be conducted. In addition, more research is needed to explore ways to transfer learned representations from the patch classifier to the whole image classifier effectively. Finally, the main limitation of this study and in general of patch-based classifiers is the need for lesion-level annotations. We emphasize the need for further research into weakly supervised learning and semi-supervised learning to minimize the reliance on annotations.

## Figures and Tables

**Figure 1 bioengineering-10-00534-f001:**
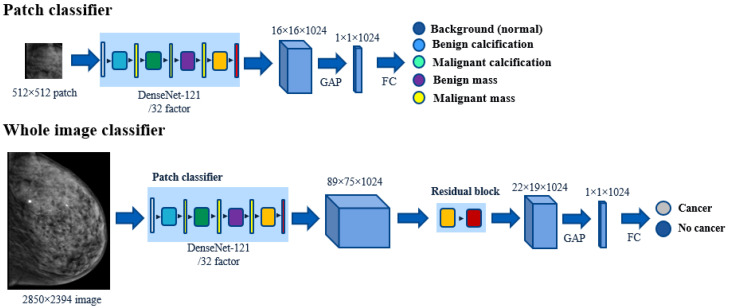
Extension of the patch classifier to the whole image classifier.

**Figure 2 bioengineering-10-00534-f002:**
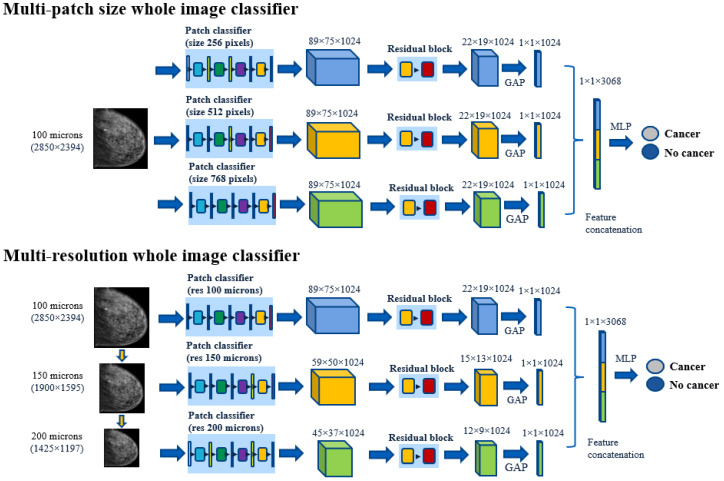
Multi-patch size and multi-resolution architectures.

**Figure 3 bioengineering-10-00534-f003:**
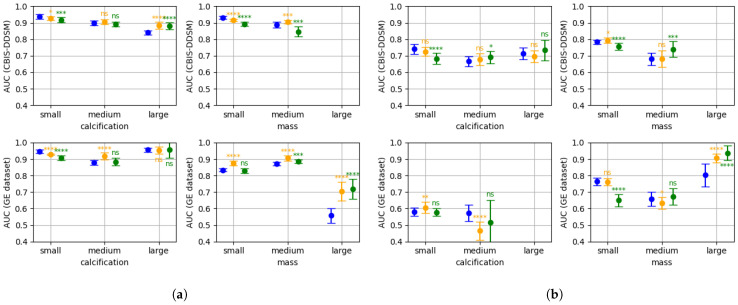
Performance of the patch classifier with different patch sizes at 100 μm resolution (10 runs): patch size 258 (blue), patch size 512 (orange), patch size 768 (green). At the top for the CBIS-DDSM and at the bottom for the GEHC data; at the left for calcification and at the right for mass. Level of statistical significance: ns (not significant), * (p<0.1), ** (p<0.05), *** (p<0.01), **** (p<0.001). Missing AUC for some lesion groups is due to an insufficient number of examples. (**a**) Normal vs. abnormal AUC. (**b**) Malignant vs. benign AUC.

**Figure 4 bioengineering-10-00534-f004:**
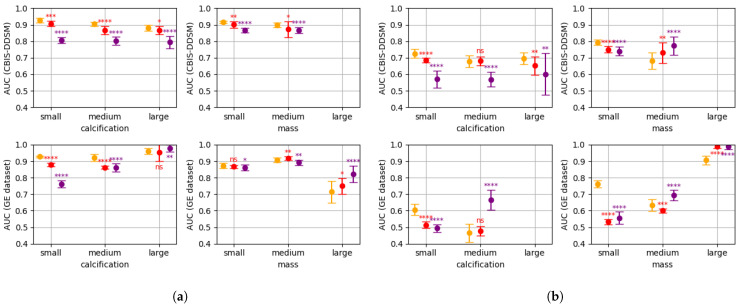
Performance of the patch classifier with different resolutions and fixed patch size of 512 pixels (10 runs): 100 μm (orange), 150 μm (red), 200 μm (violet). At the top for the CBIS-DDSM and at the bottom for the GEHC data; at the left for calcification and at the right for mass. Level of statistical significance: ns (not significant), * (p<0.1), ** (p<0.05), *** (p<0.01), **** (p<0.001). Missing AUC for some lesion groups is due to an insufficient number of examples. (**a**) Normal vs. abnormal AUC. (**b**) Malignant vs. benign AUC.

**Figure 5 bioengineering-10-00534-f005:**
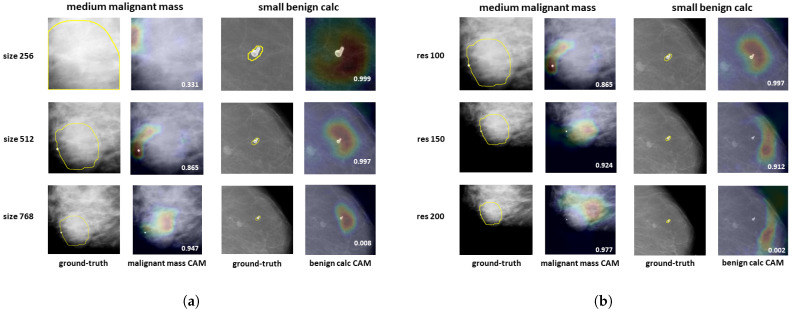
Class activation maps extracted from the last convolutional layer and for the ground-truth class at different patch sizes (**a**), and resolutions (**b**). Red and yellow areas reflect higher neuron activation. The ground-truth localization of the lesion is also shown. (**a**) Patch sizes 256, 512, and 768 pixels; (**b**) resolutions 100 μm, 150 μm, and 200 μm.

**Table 1 bioengineering-10-00534-t001:** AUC, specificity (Sp), and accuracy (Acc) of the whole image classifiers in CBIS-DDSM, with significance level (5 runs). n.a.: not applicable, n.s.: not significant. The classifiers signalled with “*” are the same models. The operating point for obtaining the specificity and the accuracy is fixed to have Se = 0.75.

	AUC	Sp (with Se = 0.75)	Acc (with Se = 0.75)
patch size 256	0.784 ± 0.002 (n.a.)	0.656 ± 0.013 (n.a.)	0.703 ± 0.014 (n.a.)
patch size 512 *	0.764 ± 0.005 (<0.001)	0.627 ± 0.029 (<0.05)	0.689 ± 0.024 (n.s.)
patch size 768	0.776 ± 0.010 (<0.1)	0.656 ± 0.023 (n.s.)	0.703 ± 0.02 (n.s.)
multi-patch size	**0.809 ± 0.005 (<0.001)**	**0.710 ± 0.020 (<0.001)**	**0.730 ± 0.019 (<0.05)**
resolution 100 *	0.764 ± 0.005 (n.a.)	0.627 ±0.029 (n.a.)	0.689 ± 0.024 (n.a.)
resolution 150	0.744 ± 0.005 (<0.001)	0.588 ± 0.013 (<0.05)	0.669 ± 0.009 (<0.1)
resolution 200	0.736 ± 0.005 (<0.001)	0.548 ± 0.029 (<0.005)	0.649 ± 0.023 (<0.05)
multi-resolution	**0.789 ± 0.005 (<0.001)**	**0.670 ± 0.005 (<0.05)**	**0.710 ± 0.006 (<0.1)**
FPN	0.788 ± 0.003 (n.a.)	0.685 ± 0.004 (n.a.)	0.717 ± 0.004 (n.a.)
multi-resolution	0.789 ± 0.005 (n.s.)	0.670 ± 0.005 (<0.001)	0.710 ± 0.006 (<0.05)
multi-patch size	**0.809 ± 0.005 (<0.001)**	**0.710 ± 0.020 (<0.05)**	**0.730 ± 0.019 (<0.1)**

**Table 2 bioengineering-10-00534-t002:** AUC, specificity (Sp), and accuracy (Acc) of the whole image classifiers in the GEHC dataset, with significance level (5 runs). n.a.: not applicable, n.s.: not significant. The classifiers signalled with “*” are the same models. The operating point for obtaining the specificity and the accuracy is fixed to have Se = 0.75.

	AUC	Sp (with Se = 0.75)	Acc (with Se = 0.75)
patch size 256	0.685 ± 0.012 (n.s.)	0.487 ± 0.023 (n.s.)	0.619 ± 0.038 (n.s.)
patch size 512 *	0.688 ± 0.011 (n.a.)	0.470 ± 0.048 (n.a.)	0.610 ± 0.041 (n.a.)
patch size 768	0.673 ± 0.010 (<0.05)	0.430 ± 0.032 (<0.1)	0.590 ± 0.033 (n.s.)
multi-patch size	**0.722 ± 0.012 (<0.005)**	**0.552 ± 0.054 (<0.05)**	**0.651 ± 0.042 (<0.1)**
resolution 100 *	0.688 ± 0.011 (n.a.)	**0.470 ± 0.048 (n.a.)**	**0.610 ± 0.041 (n.a.)**
resolution 150	0.628 ± 0.017 (<0.001)	0.333 ± 0.057 (<0.005)	0.542 ± 0.053 (<0.05)
resolution 200	0.583 ± 0.036 (<0.001)	0.333 ± 0.065 (<0.005)	0.542 ± 0.067 (<0.05)
multi-resolution	**0.709 ± 0.010 (<0.01)**	**0.466 ± 0.032 (n.s.)**	**0.608 ± 0.032 (n.s.)**
FPN	0.697 ± 0.017 (n.a.)	0.487 ± 0.095 (n.a.)	0.619 ± 0.061 (n.a.)
multi-resolution	0.709 ± 0.010 (<0.1)	0.466 ± 0.032 (n.s.)	0.608 ± 0.032 (n.s.)
multi-patch size	**0.722 ± 0.012 (<0.05)**	**0.552 ± 0.054 (<0.1)**	**0.651 ± 0.042 (<0.1)**

## Data Availability

Not applicable.

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
