# Peer review of "Exploiting Patch Sizes and Resolutions for Multi-Scale Deep Learning in Mammogram Image Classification"

_bioengineering, 2023, doi:10.3390/bioengineering10050534_

Round 1
Reviewer 1 Report
The study proposes a multi-patch-size classifier and a multi-resolution classifier for different patch sizes and resolutions for 2D mammography image classification.
I have some questions and suggestions regarding the study:
1. Feature extraction is one of the important steps in image classification. Did the authors apply feature extraction in this study? If yes, please mention the name and explain how it works in the paper. Also, please explain the parameter used for the feature extraction.
2. In this study, the performance evaluation is measured using AUC and a p-value. I am curious about the results in terms of accuracy, sensitivity, and specificity. Therefore, I suggest providing the additional results in terms of accuracy, sensitivity, and specificity utilizing 10-fold cross validation.
3. The datasets used for the study or experiment are publicly available datasets. The results of the study should be compared with those of previous studies.
4. Please add the conclusion section to the paper. It points out the summary of the main result, strength, weakness, or limitation, and the future challenge.
Reviewer 2 Report
This paper investigates the influence of image resolution and patch size on the performance for 2D mammography image classifiers, based on which a system combining multi-resolution classifier and a multi-patch-size classifier is proposed. The topic is of interest for cancer diagnosis research and the reported experimental results look promising, but the paper should be further polished before being considered for publication.
A major concern is the English writing of the paper. The authors should carefully check if there are typos or grammar errors, e.g., in line 16, does "fre diagnosed" mean to say " frequently diagnosed"? Besides, as an important line of research, a few lines of discussions should be provided to discuss the potential advantage of the proposed scheme compared with recent cancer diagnosis methods based on unsupervised learning (e.g., "Robust collaborative clustering of subjects and radiomic features for cancer prognosis" in IEEE TBME'20), especially in scenarios where annotations are rare.
Round 2
Reviewer 1 Report
All the comments and suggestions have been addressed